# Osmolality as a strong predictor of COVID-19 mortality and its possible links to other biomarkers

Sirin Cetin[1], Ayse Ulgen[2,3]*, Hakan Sivgin[4], Meryem Cetin[5], Wentian Li[6,7]*

1 Department of Biostatistics, Faculty of Medicine, Amasya University, Amasya, Türkiye, 2 Department of Mathematics, School of Science and Technology, Nottingham Trent University, Nottingham, United Kingdom, 3 Department of Biostatistics, Faculty of Medicine, Girne American University, Karmi, Cyprus, 4 Department of Internal Medicine, Faculty of Medicine, Tokat Gaziosmanpaşa University, Tokat, Türkiye, 5 Department of Medical Microbiology, Faculty of Medicine, Amasya University, Amasya, Türkiye, 6 Department of Applied Mathematics and Statistics, Stony Brook University, Stony Brook, New York, United States of America, 7 The Robert S. Boas Center for Genomics and Human Genetics, The Feinstein Institutes for Medical Research, Northwell Health, Manhasset, New York, United States of America

* ayshe.ulgen@global.t-bird.edu; wtli2012@gmail.com, wentian.li@stonybrook.edu

**Data availability statement:** The data used in this paper can be downloaded from: https://github.com/wlicol/osmolality.

## Abstract

Osmolality, concentration of solute particles, was rarely used for prognosis for COVID-19. By analyzing blood samples of more than 1300 COVID-19 patients from Tokat, Turkey (including 100 surviving and 30 deceased inpatients), we found calculated osmolality to be an excellent prognostic biomarker for mortality and significantly associated with hospitalization, independent from gender and age. Although calculated osmolality is defined as a weighted sum of sodium, glucose, and urea, the three are not necessarily independent. Other blood test biomarkers, ferritin, creatine, and chloride are also correlated with osmolality after conditioning on age. By applying a combination of collider analysis and mediation analysis, we design a pipeline to construct a causal model among all these variables in their relationship to osmolality. We confirm that while glucose and sodium are independent contributors of osmolality, glucose and urea, urea and sodium are correlated. We also conclude that ferritin and creatine are associated with osmolality through urea, and chloride's association to osmolality is through sodium.

## Introduction

Many risk factors have been identified for COVID-19 severity/hospitalization and mortality [1–3]. It is of particular interest to use routine blood test at the time of hospital admission to predict the COVID-19 disease prognosis [4,5]. The best established risk factors that are well accepted now for severity and mortality are age [6–9] and co-morbidities [10,11] such as diabetes, hypertension, and obesity [12–17].

The severity and mortality of COVID-19 is associated with a large number of risk factors. In our previous work on COVID-19 patients from Turkey, 10∼30 biomarkers are found to be associated with time-to-death or time-to-release data [18]. With a larger number of risk

**Funding:** The author(s) received no specific funding for this work.

**Competing interests:** The authors have declared that no competing interests exist.

factors, one approach is to use the top factors where the ranking is based on multiple criteria [18], and another approach is to combine multiple factors into a composite score [19].

Osmolality (either measured or calculated [20,21]) or osmolarity are measure of the number of osmotically active molecules dissolved in the blood (per unit weight or per unit volume). Some of these molecules are the by-products (waste) of certain metabolic process, and their level may reflect biochemical abnormality. There are several formulas for calculated osmolality [22,23]. The formula we use here is a simple linear combination of three measurements: sodium, urea (a nitrogenous breakdown product), and glucose [24] plasma osmolality = $2 \times$ sodium + urea/5.99 + glucose/18. An alternative of this formula is: plasma osmolality = 2 (Sodium) + (BUN)/2.8 + (Glucose)/18, but BUN (blood urea nitrogen) is approximately half (28/60 = 1/2.14) of level of urea, because of nitrogen content in urea has a molecular weight of 28 out of the total molecular weight of 60. Other similar formula can be found in [25,26]. In this paper, we will use term osmolality to imply the calculated osmolality. Sodium apparently contributes the most numerically to osmolality (96.5% in our data), followed by urea (1.9%) and glucose (1.5%).

All components of osmolality – sodium, urea, and glucose, are known to be significant biomarkers for COVID-19 severity/mortality [18]. Our previous study to rank factors from time-to-death, time-to-release, and both, result in ranks of #1, #11, #4 for urea, #8, #12, #9 for glucose, and #9, #20, #15 for sodium [18]. Other publications have also pointed out links between COVID-19 disease severity and hypernatremia/hyponatremia, high urea and hyperglycemia (e.g., [27–32]). It is reasonable to assume that osmolality itself could be an important risk factor for COVID-19. Osmolality had been proposed as a prognosis biomarker for other diseases such as stroke [33], heart failure [34–37], pulmonary embolism [38]. However, its application to COVID-19 patients was limited in the literature (see, e.g., [39,40]).

In this paper, we first establish that osmolality is one of the best prognostic biomarkers for COVID-19. We then use osmolality as the focus of our attention to link several other factors that are also associated with COVID-19 severity. At a first glance, as sodium, urea, and glucose are the only components in the definition of calculated osmolality, there could be no "indirect path", using a jargon from the mediation analysis, to be explained in the Method section, from other factors to be involved in osmolality. In order to untangle the links among a set of correlated variables, we design a pipeline utilizing collider analysis (see, e.g., [41] and Sect 6.2 of [42]) and mediation analysis (see, e.g., [43,44]) (both will be discussed in the Method section), to assign causal arrows for any three-variable at a time. Using the pipeline, we first decide the relationship between the three components for osmolality: sodium, urea, and glucose. Then we decide which of these three components are more likely to be linked to other variables that contribute to COVID-19 severity: ferritin, creatine, and chloride, by another round of collider analysis and mediation analysis as described in the pipeline.

## Data and methods

### COVID-19 patient data

The COVID-19 patient data was collected from Tokat State Hospital, Tokat, Turkey, from March 30, 2020 to December 1, 2020. The data were accessed from 01/05/2021 to 01/07/2022 for research purposes. There are n = 1323 patients, with 364 male and 959 female. The median age is 41. The COVID-19 status was determined by nasopharyngeal swab samples tested by RT-PCR. Inpatients were all admitted to hospital according to the WHO confirmation guideline. Subjects were fully anonymized before we accessed them. Verbal consent was informed from the patients at the medical visit. No minors were recruited in the study. This is a retrospective study and all data are fully anonymized before accessing them.

The following patients were excluded from this dataset: 1. those with hematological malignancies, immunodeficiency states, chronic kidney disease, hearth failure and those on renal replacement therapy; 2. those who received osmotic agents like mannitol prior to the first blood draw with RT-PCR confirmed in hospitalized COVID-19 patients from the study; 3. those with age 17 or younger.

As a retrospective study, the patient data were extracted from the medical case records. The blood samples were collected within 24 hours of hospital visit/admission. This study was carried out with the approval of Tokat Gaziosmanpasa University Faculty of Medicine Non-Interventional Clinical Research Ethics Committee (decision No: 21-KAEK-104)

The following blood test factors are used in this paper: urea, ferritin, creatine, calcium, sodium, chloride, potassium, glucose, vitamin D, lactate dehydrogenase (LDH), C-reactive protein (CRP), aspartate aminotransferase (AST), alanine aminotransferase (ALT). There were more blood test factors that are not included here because of a large proportion of missing data, in particular, various immune cell counts (missing rates more than 80%), D-dimer (missing rate 59%), activated partial thromboplastin time (APTT, missing rate 47%), and fibrinogen (37%). In this study, all patients had routine blood samples taken, but approximately 5 = 11% of records in COVID-19 positive patients were missing LDH, CRP, and chloride results, due to unplanned gaps from changes in admission protocols, variation in clinical priorities, or retrospective record limitations. But importantly, test results of the three components of osmolality (sodium, potassium, glucose) – our primary target of analysis, are fully available. Although the SARS-CoV-2 variant was not tested in these samples, the dominating variant in Turkey in 2020 is alpha variant [45].

The data used in this paper can be downloaded from https://github.com/wlicol/osmolality.

## Potential confounding factors not collected in our data

We do not have SARS-CoV-2 variant information for individual patients. The B.1.1.7 (alpha) variant was first reported in the UK in November 2020 and was known to spread fast in mid-December 2020 in UK [46,47] and elsewhere. In an independent investigation, two strains of SARS-CoV-2 seemed to dominate Turkish cases in 2020 [48]: the first was GISAID clade L which contains the original reference genome, and the second was GISAID clade GRY which is the alpha variant [48]. Because alpha variant increased disease severity risk either overall or within certain age range [49–51], a patient's variant status has potential to impact the analysis result.

However, we have only 4 inpatients (out of 130, or 3%) were collected in November-December 2020 period, and one of the 6 ICU patients were from the same period. Although this enrichment from inpatient to ICU may indicate an effect of the possible alpha variant at work, the overall number of samples involved is very small. The implication is that our result is very unlikely to be affected by having the variant information, as most would have the original strain of SARS-CoV-2.

All patients collected in our data, both inpatients and outpatients, were treated with favipravir antiviral medicine. All inpatients were treated by attending physicians according to the current national guidelines prepared by the Scientific Advisory Committee of the Turkish Ministry of Health. There were no biased treatments for inpatients that may impact the disease outcome. Therefore, we do not consider treatment status as a potential confounding factor.

Finally, COVID-19 vaccination has been proven extremely effective in preventing disease severity. Vaccination status is clearly a confounding factor. However, the first administered COVID-19 vaccine shot in Turkey was on January 13, 2021, and for health care workers only

(and not available for the general public). Since all our patients were collected in 2020, none of them were vaccinated.

## Collider analysis and conditional analysis

We use partial/conditional correlation to measure correlation between two continuous variables (X and Y) after controlling the effect of the third variable (Z). We may use the notation cor(X,Y| Z) for partial or conditional correlation. In causal graphs, if two independent causes (X and Y) collide on the outcome variable Z, we expect cor(X,Y) = 0 and cox(X,Y|Z) $\neq$ 0 (often negative). The induced correlation between X and Y by conditioning on Z is the key part of a collider analysis.

## Mediation analysis

Mediation analysis deals with the following question: if the independent variable X contributes to outcome variable Y both directly, or independently through a third mediation variable M, how to estimate the proportion of contributions in the two causal paths [44]. The classic mediation analysis assumes all three variables to be continuous, and the relationship between them can be summarized by three linear regressions (two of them are independent) [44]. There is also a more modern "causal mediation analysis" that can handle a variety of variable types [52]. A mediation analysis program would estimate the proportion ($p_M$) of contribution from $X$ to $Y$ through the indirect path, $X \rightarrow M \rightarrow Y$, the confidence intervals of $p_M$, and $p$-value for testing the null hypothesis that $p_M = 0$. The direct path $X \rightarrow Y$, is the one where the impact of $X$ on $Y$ does not go through $M$, and its proportion is $1 - p_M$.

## Logistic regression

Four logistic regressions were carried out for each factor $x$ with either mortality or hospitalization as the binary outcome $y$: (reg1) raw data $y \sim x$; (reg2) logistic regression using standardized $x' = (x - mean(x))/sd(x)$ (where sd stands for standard deviation): $y \sim x'$; (reg3) conditional on age and gender $y \sim x + age + gender$; (reg4) logistic regression using log-transform and standardized $x'' = (\log(x) - mean(\log(x)))/sd(\log(x))$: $y \sim x''$.

Logistic regressions model the log-odds as a linear function of $x$, where odds is defined as prob(y = 1)/(1-prob(y = 1)). Odds-ratio (OR) is a ratio of two odds evaluated at one unit apart, i.e., odds at $x_0 + 1$ and at $x_0$ (for reg1 and reg3). In reg2 model, since the independent variable is standardized, OR is the ratio of odds at $x_0 + sd(x)$ and at $x_0$. In reg4 model, OR is the ratio of odds at $\log(x_0) + sd(\log(x))$ and at $\log(x_0)$, or equivalently, odds at $e^{sd(\log(x))} \cdot x_0$ and at $x_0$.

## ROC analysis

ROC is the abbreviation for receiver operating characteristic) [53,54]. When a continuous factor is used to predict or classify a medical outcome, a cutoff (a threshold) is chosen, partition the samples into a 2-by-2 count table, two correct predictions (true positives and true negatives) and two errors (false positives and false negatives)[55]. In ROC, the $x$-axis is the false positive rate (false positive counts divided by total number of negatives) and $y$-axis being true positive rate (true positive counts divided by total number of positives). The point ($x,y$) = (0,0) represents the most stringent threshold where all samples are called negative; whereas the point ($x,y$) = (1,1) represents the case where the threshold is so low that all samples are called positive. Gradually relaxing the threshold for calling positive results in a curve in a ROC plot (ROC curve). The area under a ROC curve is abbreviated as AUC. If a biomarker

lacks any prognostic ability, the curve is simply the diagonal line and AUC is 0.5. A larger AUC is an indication of a better prognostic marker.

### R packages

The R package *ppcor* is used to calculate the partial correlation [56]. Conditional or correlation analysis have been used in other projects of ours in the past [57,58].

The R package CMAverse [59] (*bs1125.github.io/CMAverse/* ) is used for causal mediation analysis. The function *cmest* is used and the pm ($p_M$: proportion of indirect impact) value, its confidence interval, and *p*-values, is reported.

The R package *pROC* [60] is used to calculate AUC, area under a ROC curve.

For ordinal regression, we use the *polr* function from MASS package [61] (*cran.r-project.org/web/packages/MASS/*), and function *coeftest* from the AER package [62] (*cran.r-project.org/web/packages/AER/*) to get *p*-value from a polr object.

Besides the ppcor, CMAverse, pROC, MASS, and AER, we use R (*www.r-project.org*) for other standard statistical analyses. *lm* is used for linear regression, *glm( y ∼ x, family = "binomial")* for logistic regression (and *glm( y ∼ x + gender + age, family = "binomial")* for multiple regression with gender and age as covariates), *cor(x,y, method = "spearman")* for Spearman correlation coefficient.

A lower *p*-value implies that the data is unlikely to be produced from a null hypothesis (e.g., no correlation, regression coefficient is zero, etc.). How small a *p*-value can we claim the test result to be "significant"? We follow the argument in [63–65] to use the *p*-value = 0.001 or 0.005 as the cutoff threshold.

## Results

### Osmolality is one of the best prognostic biomarkers for COVID-19 mortality

Of the n = 1323 patients, n = 1193 were outpatients and n = 130 were hospitalized (including 6 in ICU). Then, out of n = 1323 patients, n = 30 deceased (one of them died in ICU). Because the number of samples in ICU is too small (less power for a statistical analysis), we focused on deceased or hospitalized patients as the contracting group with the rest.

We test a series of logistic regressions for outcome (death, or hospitalization, or both) by using the original value, using the standardized value (which wouldn't affect the *p*-value, but affect the regression coefficient, thus odds-ratio), adding gender and age as covariates (which may make a factor less significant as a risk factor because age may take away some of the risk signal), using log-transformed values (because many factors are better normally distributed only after a log transformation).

Since reg1,reg3,reg4 all have different meanings for OR, the definition of OR depends on the scale as well as the distribution of the variable. It would be inappropriate to log transform a factor, and use the corresponding OR, if it actually follows a normal distribution; on the other hand, it is inappropriate to use the raw data if the factor follows a log-normal distribution. Since the factors in our data follow different distributions, it is unclear which OR (from which version of logistic regression) should be used to compare the effect size of different factors. To deal with this problem, we use the AUC of ROC as a measure of effect size.

*P*-value is a measure of statistical significance and is not comparable between studies with different sample sizes. However, when comparing different factors on the same outcome, *p*-values are comparable because the sample size is the same (except for some disadvantage

for LDH, chloride, and CRP as these have slightly lesser samples). AUC as a measure of prediction performance is comparable among factors. Therefore, we use the *p*-value from reg1 (raw data), from reg4 (log-transformed, plus conditional on age and gender), and from AUC of ROC, to construct three ranking lists of these factors. The three ranks are averaged which is used to organize Table 1 (smaller *p*-values and/or larger AUC are on the top). The top five factors (all have AUC larger than 90%) are urea, calcium, ferritin, osmolality, and LDH. The next group, all have AUC between 70% and high 80%s, includes creatine, sodium, glucose, chloride, AST, CRP, and vitamin D. The last two factors, potassium and ALT, are not good prognostic markers for mortality.

Table 1 also shows that for many factors, reg4 is less significant than reg1. One important difference between reg4 and reg1 is that reg4 adjusts the contribution from the age variable. One may treat age as a surrogate to deterioration of body at cellular and systemic level. On the

**Table 1**. Measuring and Testing a factor's association with mortality by logistic regression and AUC.

| factor | n(NA) (rate) | range | reg1 raw OR(95%CI) *p*-value | reg2 standardize OR (95%CI) | reg3 gender + age OR(95%CI) *p*-value | reg4 log,sd,gender + age OR(95%CI) *p*-value | ROC AUC (95%CI) | rank |
|---|---|---|---|---|---|---|---|---|
| age | | (18 -90) | 1.16 (1.11-1.2) **7.8E-14** | 10.3 (5.59-18.99) | NA | NA | NA | |
| urea | 0 | (8.84 -296.7) | 1.04(1.03-1.05) **5.2E-18** | 2.55 (2.06-3.15) | 1.02(1.015-1.03) **3.4E-7** | 3.18(2.21-4.59) **5.7E-10** | 0.97 (0.94-1) | 1 |
| calcium | 5 (0.4%) | (5.97 -11.9) | 0.06(0.032-0.11) **2.9E-19** | 0.169 (0.12-0.25) | 0.14(0.075-0.26) **6.5E-10** | 0.34(0.24-0.48) **7.1E-10** | 0.96 (0.93-0.99) | 2 |
| ferritin | 26 (2%) | (1 -2000) | 1.003(1.003-1.004) **1.2E-19** | 2.31 (1.93-2.77) | 1.003(1.002-1.004) **9.1E-10** | 7.86(3.91-15.79) **6.9E-9** | 0.95 (0.91-0.98) | 3 |
| osmolality | 0 | (259.8 -367.3) | 1.13(1.1-1.16) **1E-19** | 3.25 (2.52-4.2) | 1.08(1.05-1.11) **1.4E-8** | 2.16(1.66-2.81) **8.7E-9** | 0.92 (0.85-0.99) | 4 |
| LDH | 168 (10.6%) | (98 -2701.2) | 1.012(1.009-1.014) **8E-18** | 4.65 (3.28-6.6) | 1.009(1.006-1.014) **2.4E-9** | 3.55(2.33-5.41) **4.2E-9** | 0.94 (0.91-0.98) | 5 |
| creatine | 0 | (0.3 -8.7) | 5.24(2.43-8.01) **2.1E-14** | 2.1 (1.74-2.54) | 2.64(1.66-4.21) **4.4E-5** | 2.01(1.55-2.6) **9.8E-8** | 0.86 (0.75-0.96) | 6 |
| sodium | 0 | (121.9 -165.8) | 1.39(1.28-1.51) **1.5E-14** | 2.93 (2.23-3.86) | 1.2(1.11-1.29) **4.5E-6** | 1.83(1.41-2.38) **5.3E-6** | 0.77 (0.64-0.90) | 7 |
| glucose | 0 | (35.9 -627.3) | 1.01(1.007-1.013) **1.9E-10** | 1.72 (1.46-2.03) | 1.007(1.004-1.011) **8.3E-5** | 1.73(1.31-2.28) **1.1E-4** | 0.88 (0.81-0.94) | 8 |
| chloride | 122 (9.2%) | (87.5 -137.3) | 1.3(1.2-1.4) **7.5E-10** | 2.64 (1.94-3.59) | 1.19(1.1-1.29) **8.5E-6** | 1.98(1.47-2.66) **6.9E-6** | 0.70 (0.57-0.84) | 9 |
| AST | 0 | (1.4 -504.5) | 1.02(1.015-1.03) **7.3E-8** | 1.57 (1.33-1.85) | 1.03(1.02-1.04) **5.3E-5** | 2(1.47-2.73) **1E-5** | 0.75 (0.64-0.86) | 10 |
| CRP | 179 (11.8%) | (0.1 -452) | 1.02(1.01-1.03) **4.5E-7** | 1.71 (1.39-2.11) | 1.01(1.004-1.02) 0.0038 | 2.21(1.46-3.34) **1.4E-4** | 0.80 (0.71-0.90) | 11 |
| vitaminD | 0 | (2 -93.3) | 0.84(0.78-0.91) **2.5E-5** | 0.15 (0.063-0.36) | 0.85(0.78-0.92) **9.2E-5** | 0.4 (0.26-0.61) **1.8E-5** | 0.77 (0.69-0.84) | 12 |
| potassium | 3 (0.2%) | (2.8 -6.27) | 0.72(0.31-1.72) 0.47 | 0.87 (0.6-1.26) | 0.66(0.34-1.29) 0.22 | 0.77(0.58-1.03) 0.079 | 0.59 (0.43-0.74) | 13 |
| ALT | 0 | (1.4 -504.5) | 1.006(1.001-1.01) 0.012 | 1.23 (1.05-1.44) | 1.007(1.001-1.013) 0.032 | 1.1(0.79-1.56) 0.55 | 0.54 (0.41-0.67) | 14 |

n(NA): number of samples without the factor's information (NA for "not available"); The missing rate below is n(NA) divided by n = 1323. range: numerical range of the factor's value; reg1: odds-ratio (OR) and 95% confidence interval (CI) and *p*-value in logistic regression model 1 (single-variable without any transformation); reg2: those results for model 2 (variables are standardized to have mean = 0, variance = 1); reg3: results for model 3 (single-variable without transformation, but with gender and age as two co-variates); reg4: results for model 4 (variable is log-transformed, then standardized, plus covariates of gender and age); AUC: area under curve of ROC (receiver operating characteristic); rank: based on the sum of three ranks from *p*-value in reg1, *p*-value in reg4, and AUC. In these three ranking orders, the number 1 rank is given to the factor with the smallest *p*-value, or the largest AUC. A smaller than 1E-3 = 0.001 *p*-value is marked in bold.

other hand, even after an adjustment of age, many factors are still significant, which indicates that the general aging process is not the only reason that some patients have higher risk for mortality.

We also noticed from Table 1 that osmolality is ranked no.4, behind its component urea, but above its components sodium and glucose. Judging from the *p*-value for reg1/reg2 and reg3, osmolality is even better than urea. Whether osmolality is slightly below or above urea, we argue that osmolality is probably more robust than urea as a prognostic marker because it contains more information.

## Osmolality is a good prognostic biomarker for COVID-19 hospitalization

Table 2 shows similar four logistic regressions as the last subsection, but with hospitalization as the outcome variable. The order of the factor is identical to that in Table 1. However, the last column also shows the ranking order with hospitalization (H) as outcome (average over the ranking from reg1, reg4, and AUC), plus reproducing the ranking order in Table 1 (D for death), as well as the average ranking order based on six ordering lists (three for mortality, three for hospitalization) (A for all). The osmolality is ranked no.4 in all three lists. A non-integer rank indicates a tie.

Interestingly, for hospitalization outcome, as well as for overall mortality and hospitalization, osmolality is ranked higher than all of its components (urea, sodium, glucose). Again, we speculate that it is due to the fact that osmolality contains more physiological information than each of its components.

Similar to Table 1, adjusting for gender and age may make an originally significant risk factor to be non-significant (e.g. glucose), or greatly reduce its significance. At the *p*-value of 0.001 level, only three factors are significant for age-conditioned (plus log-transformed) regression analysis: LDH, CRP, and calcium. Generally speaking, our factors are better prognostic markers for mortality than for hospitalization, which include the mortality as a subset.

## Combining mortality and hospitalization by ordinal logistic regression

To combine the results from the previous subsections on mortality and hospitalization, one may also treat them as two level of severity of different degrees. Although it is possible to code outpatients, surviving inpatients and deceased inpatients as 0,1,2, and apply a linear regression, a better option is to use the ordinal regression where 0,1,2 are ordered factor levels, not numerical values.

Table 3 shows the *p*-value of ordinal logistic regression in three versions: raw data (reg1), raw data with age and gender as covariate (reg3), logarithm-transformed, then standardized factor value with age and gender as covariate (reg4). The second version of ordinal logistic regression: standardized factor value, produces the same (or essentially the same) *p*-value as reg1.

The result in Table 3 is consistent with that in Table 1 and Table 2, with calcium, LDH, ferritin, osmolality as the most significant factors for COVID-19 prognosis. CRP is more significant for hospitalization than mortality, whereas urea is more significant for mortality than hospitalization. Interestingly, two common known risk factors, vitamin D and glucose, are near the bottom of list: glucose loses its significance mostly by conditioning on age and vitamin D is only borderline significant.

**Table 2. Measuring and testing a factor's association with hospitalization by logistic regression and AUC.**

| factor | reg1 | reg2 | reg3 | reg4 | ROC | rank |
|---|---|---|---|---|---|---|
| | raw | standardize | gender + age | log,sd,gender + age | | |
| | OR(95%CI) | OR(95%CI) | OR(95%CI) | OR(95%CI) | AUC | (D,H,A) |
| | p-value | | p-value | p-value | (95%CI) | |
| urea | 1.028(1.02-1.035) | 1.92(1.62-2.27) | 1.009(1.002-1.016) | 1.25(1.02-1.51) | 0.68 | (1,6,5) |
| | **7.4E-14** | | 0.011 | 0.028 | (0.63-0.73) | |
| calcium | 0.27(0.2-0.36) | 0.44(0.37-0.52) | 0.53(0.4-0.71) | 0.68(0.57-0.81) | 0.70 | (2,2,1) |
| | **4.5E-20** | | **1.8E-5** | **1.5E-5** | (0.65-0.76) | |
| ferritin | 1.003(1.002-1.003) | 1.89(1.63-2.2) | 1.001(1.001-1.002) | 1.44(1.12-1.85) | 0.71 | (3,3,3) |
| | **1E-16** | | **3.8E-6** | 0.0041 | (0.66-0.76) | |
| osmolality | 1.08(1.06-1.10) | 2.06(1.74-2.43) | 1.03(1.01-1.05) | 1.3(1.1-1.54) | 0.68 | (4,4,4) |
| | **4.3E-17** | | 0.0018 | 0.0017 | (0.63-0.74) | |
| LDH | 1.008(1.006-1.009) | 2.75(2.19-3.46) | 1.004(1.002-1.006) | 1.56(1.29-1.9) | 0.74 | (5,1,2) |
| | **3.9E-18** | | **4.5E-5** | **6E-6** | (0.69-0.79) | |
| creatine | 3(2.12-4.26) | 1.64(1.4-1.91) | 1.45(1.05-2.02) | 1.21(1.03-1.42) | 0.62 | (6,8,6) |
| | **6E-10** | | 0.026 | 0.023 | (0.56-0.67) | |
| sodium | 1.18(1.12-1.24) | 1.71(1.45-2.02) | 1.07(1.02-1.12) | 1.23(1.05-1.45) | 0.61 | (7,7,7) |
| | **3.1E-10** | | 0.0083 | 0.0098 | (0.55-0.68) | |
| glucose | 1.006(1.004-1.009) | 1.43(1.26-1.62) | 1.001(0.998-1.004) | 1.07(0.9-1.26) | 0.67 | (8,9.5,9) |
| | **5.1E-8** | | 0.42 | 0.46 | (0.62-0.72) | |
| chloride | 1.01(0.96-1.06) | 1.04(0.86-1.24) | 1.02(0.98-1.06) | 1.06(0.91-1.25) | 0.53 | (9,12.5,12) |
| | 0.7 | | 0.39 | 0.46 | (0.47-0.6) | |
| AST | 1.016(1.009-1.023) | 1.36(1.19-1.56) | 1.01(1-1.02) | 1.21(1.02-1.43) | 0.6 | (10,9.5,10) |
| | **1.2E-5** | | 0.0069 | 0.033 | (0.54-0.66) | |
| CRP | 1.02(1.013-1.026) | 1.66(1.39-1.99) | 1.01(1-1.02) | 1.63(1.31-2.03) | 0.69 | (11,5,8) |
| | **2.1E-8** | | 0.0041 | **1.2E-5** | (0.64-0.74) | |
| vitaminD | 0.98(0.96-1) | 0.83(0.67-1.02) | 0.98(0.96-1) | 0.81(0.67-0.99) | 0.55 | (12,11,11) |
| | 0.08 | | 0.055 | 0.037 | (0.5-0.61) | |
| potassium | 1.14(0.75-1.72) | 1.06(0.88-1.26) | 0.78(0.52-1.16) | 0.88(0.74-1.04) | 0.5 | (13,12.5,13) |
| | 0.55 | | 0.21 | 0.14 | (0.44-0.56) | |
| ALT | 1.003(0.999-1.006) | 1.1(0.96-1.26) | 1.002(0.997-1.006) | 0.97(0.8-1.18) | 0.53 | (14,14,14) |
| | 0.16 | | 0.51 | 0.76 | (0.47-0.59) | |

Similar to Table 1, but changing the outcome variable from death/mortality to hospitalization status. A p-value is in bold if it is smaller than 1E-3 = 0.001. The order of factors is identical to that in Table 1 (under D of the last column); however, we also list the ranking in terms of association with hospitalization (under H), and ranking from both mortality and hospitalization (under A). The H-ranking is based on the p-value from reg1, that from reg4, and AUC of ROC. If three ranking lists lead to a tie for H-ranking, a fractional rank is used (e.g., if two factors are tied at no.9, they are assigned to a rank of 9.5).

## Factors correlated with osmolality besides its components, either directly or after conditional on age

Since sodium, urea, and glucose are components of calculated osmolality, it is not surprising that these three quantities are highly correlated with osmolality (Table 4). The reason we also show results using outpatients only in Table 4 is to avoid the potential selection bias, which would be present if both a factor and osmolality are causes of the disease severity. Besides these, ferritin, creatine, ALT, LDH, CRP, AST, chloride are also significantly correlated with osmolality (though chloride loses the significance at 0.001 level if only outpatients are used). On the other hand, potassium, calcium, and vitamin D are not significantly associated with osmolality.

We use Spearman correlation in this calculation so that it is not required that these variables follow a normal distribution (another choice is to use the Pearson correlation on log-transformed variable values).

**Table 3. P-values for testing a factor's association with both hospitalization and mortality by ordered/ordinal logistic regression.**

| factor | *p*-value | | |
| --- | --- | --- | --- |
| | reg1 | reg3 | reg4 |
| | raw | gender + age | log,sd,gender + age |
| calcium | **4.6E-26** | **1.3E-9** | **4.6E-10** |
| LDH | **6.4E-23** | **8.1E-9** | **3.7E-10** |
| ferritin | **2.3E-23** | **1.5E-10** | **2.5E-5** |
| osmolality | **0** | **9.2E-8** | **5.1E-8** |
| urea | **6.4E-20** | **2.7E-6** | **1.5E-5** |
| CRP | **3.2E-10** | **9.5E-5** | **4.7E-7** |
| sodium | **3.6E-13** | **3.4E-5** | **5.4E-5** |
| creatine | **8.4E-14** | **0.00014** | **6.3E-5** |
| AST | **1.9E-7** | **3.7E-5** | **0.00089** |
| vitaminD | 0.057 | 0.0095 | 0.003 |
| chlorine | 0.4 | 0.013 | 0.021 |
| glucose | **6.2E-9** | 0.091 | 0.1 |
| potassium | 0.57 | 0.16 | 0.073 |
| ALT | 0.12 | 0.2 | 0.96 |

Similar to Tables 1 and 2, but changing the outcome variable to an ordered categorical variable representing three levels of COVID-19 prognosis (outpatient, surviving inpatient, deceased inpatients). The second regression (reg2) using the standardized factor value either lead to an identical (or similar) *p*-value as the first regression (reg1) using the raw data, thus not included in the table. The factors are ranked by the mean of logarithm of *p*-values of the last two columns, from small (more significant) to large (not significant).

**Table 4. Spearman correlation (scc) between individual factors and osmolality.**

| factor | outpatients | | all patients | | all (cond. age) | |
| --- | --- | --- | --- | --- | --- | --- |
| | scc | pv | scc | pv | scc | pv |
| sodium (Na) | 0.83 | **4.3E-308** | 0.83 | **0** | 0.85 | **0** |
| urea | 0.44 | **4.2E-57** | 0.46 | **2.4E-70** | 0.3 | **2E-28** |
| age | 0.42 | **7.5E-52** | 0.46 | **5.2E-69** | NA | NA |
| glucose | 0.35 | **1E-35** | 0.39 | **3.9E-48** | 0.22 | **8E-16** |
| ferritin | 0.3 | **7.5E-52** | 0.46 | **5.2E-69** | 0.3 | **4.9E-29** |
| creatine | 0.27 | **1.2E-21** | 0.29 | **9.9E-27** | 0.17 | **1E-19** |
| ALT | 0.2 | **5.9E-12** | 0.18 | **1.7E-11** | 0.057 | 0.04 |
| LDH | 0.16 | **2.4E-7** | 0.23 | **1.3E-15** | 0.064 | 0.027 |
| CRP | 0.11 | **0.00022** | 0.15 | **5.6E-7** | 0.05 | 0.12 |
| AST | 0.099 | **0.00063** | 0.12 | **1.1E-11** | 0.014 | 0.61 |
| chloride | 0.094 | 0.0021 | 0.12 | **2.3E-5** | 0.23 | **2.2E-15** |
| potassium (K) | 0.064 | 0.028 | 0.051 | 0.066 | -0.03 | 0.28 |
| calcium (Ca) | −0.007 | 0.81 | −0.08 | 0.0038 | 0.001 | 0.72 |
| vitaminD | 0.05 | 0.08 | 0.003 | 0.91 | -0.04 | 0.13 |

outpatients: only the outpatients are used; all patients: all patients are used; all(cond. age): using all patients but scc is conditional (partial correlation) on age. The *p*-value is for testing scc = 0, and those *p*-values smaller than 0.001 are marked with bold font.

The correlation between osmolality and other factors can be viewed by scatter plot (Fig 1). Deceased patients are labeled with red, hospitalized patients are labeled with blue, and the outpatients are in grey. From Fig 1, it can be seen directly that osmolality is high in many deceased patients (roughly 20 out of 30).

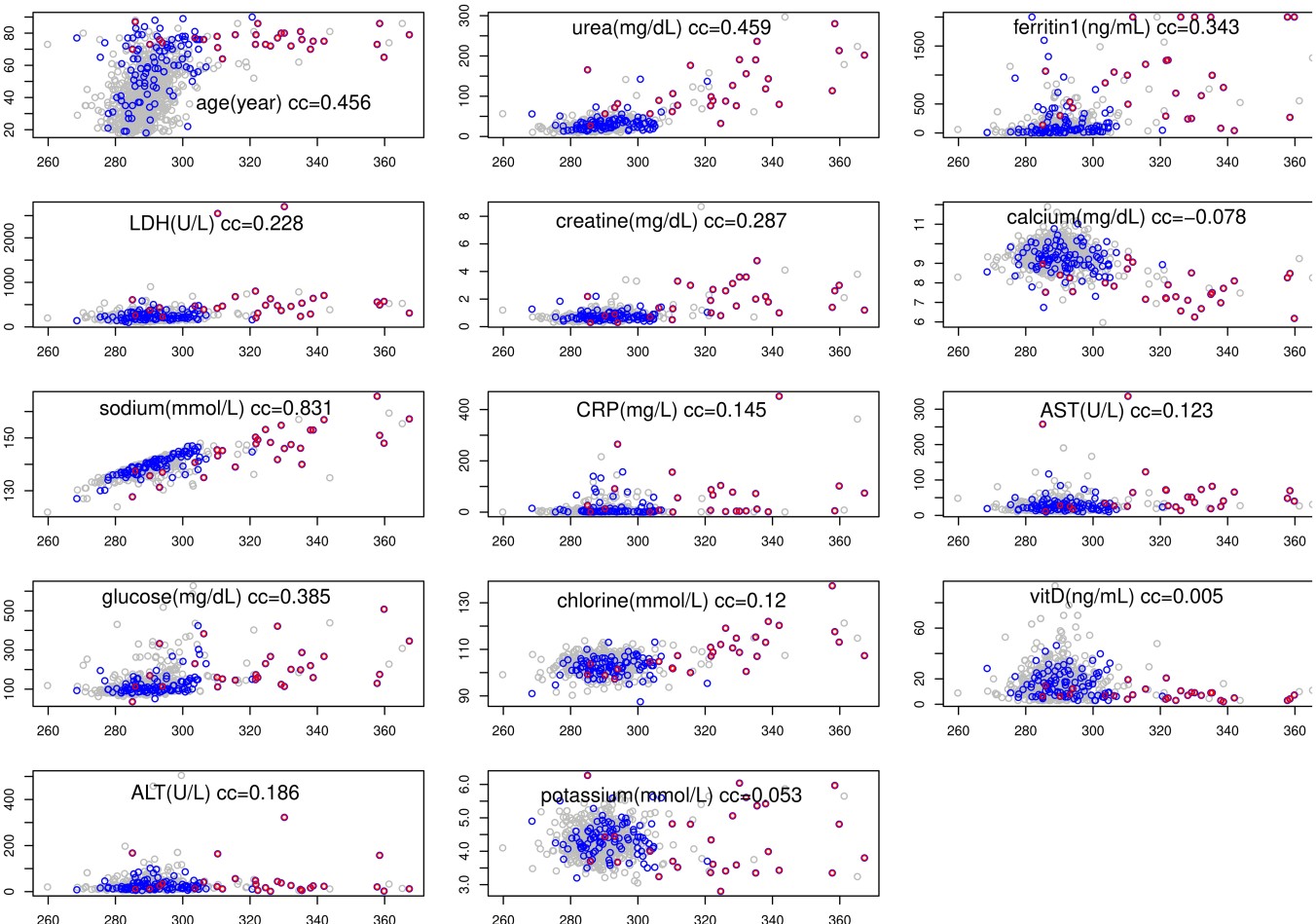

**Fig 1. Scatter plots of a factor (y-axis) versus osmolality (x-axis).** Outpatients, surviving inpatients, and deceased patients are marked with grey, blue, and red color. The cc is the Spearman correlation coefficient between the factor and osmolality calculated from all patients (see Table 4). The unit of osmolality is mOsm/kg or mOsm/L, and units for other factors are given in the plots.

Since a correlation between two variables might be due to the fact that a third variable (confounder) is a cause of both variables. To exclude this possibility, we re-calculate the correlation between a factor and osmolality conditional on the age variable. For conditional correlation, only ferritin, creatine, and chloride are significant, whereas ALT, LDH, CRP, and AST lose their significance.

## Pipeline for constructing a causal model for osmolality components and other associated test variables

Fig 2 shows our plan in proposing a causal model between osmolality and its components, as well as other associated variables. In the first step, all three components (glucose, urea, sodium) should have an arrow pointing to osmolality, simply by definition. However, it is unclear if there should be arrows among these components. Towards that, we pick two of the three components as $X$ and $Z$, osmolality as $Y$, and use the collider analysis

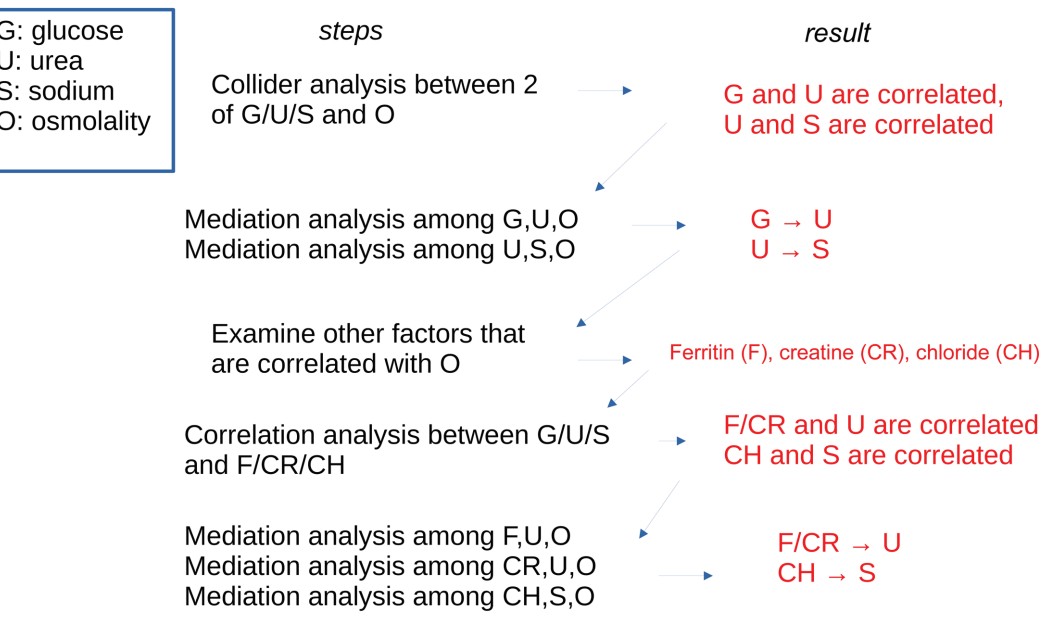

**Fig 2. Pipeline for constructing a possible causal model between osmolality and its components as well as other associated variables.**

(correlation between $X$ and $Z$, with or without conditional on $Y$ [66]) to determine if the correlation between $X$ and $Z$ are completely due to their contribution to $Y$. If yes, there is no arrow between $X$ and $Z$.

If two of the three components of osmolality are not independent, i.e., there is a causal arrow between the two, there is still a question of the direction of the arrow. For that we use mediation analysis. If the indirect path $X \to Z \to Y$ has a stronger evidence than the indirect path in another model, $Z \to X \to Y$, then we assign an arrow from $X$ to $Z$.

Next, we bring in new variables to the causal model. These are the factors/variables that strongly contribute to the COVID-19 severity, but not part of the official components of osmolality. Let's denote one of these new variables $Z^*$. We check which one of three components of osmolality is most correlated with a $Z^*$: that component should be paired with that $Z^*$.

With the result from the above collider analysis, we assign the $Y^*$ to an osmolality component $X$ if these are not independent. The last step would be the determination of an arrow direction between $X$ and $Y^*$. This can in principle be decided by a mediation analysis between $X$, $Y^*$, and $Y =$ osmolality. But as we will see later, the result is obvious without the need for analysis.

## Possible relationship between the three components of osmolality: sodium, urea, and glucose

Table 5 shows the conditional analysis (collider analysis) of any two components of osmolality and osmolality itself. Glucose and sodium are uncorrelated, but become correlated after conditioning on osmolality. This is consistent with the causal model that glucose and sodium both have a causal arrow pointing to osmolality, but there is no arrows between glucose and sodium. A negative correlation after conditioning on the collider is a common result: if the outcome $Y$ receives a contribution from $X$, the contribution from $Z$ becomes unnecessary,

**Table 5. Collider analysis of the three components of osmolality.**

| pair | cor (Spearman) | conditional cor (Spearman) |
|---|---|---|
| glucose and urea | 0.269 (pv = 7E-44) | 0.11 (pv = 5E-5) |
| urea and sodium | 0.136 (pv = 7E-7) | −0.5 (pv = 3E-84) |
| glucose and sodium | 0.05 (pv = 0.07) | −0.53 (pv = 7E-95) |

cor(Spearman): Spearman correlation between a pair of components of osmolality; conditional cor(Spearman): same Spearman correlation between two components of osmolality conditional on osmolality.

and vice versa. Note that since our continuous variables tend to be not normally distributed, a calculation of correlation is done by the non-parametric version of Spearman correlation.

The glucose-urea pair (and urea-sodium pair to a lesser extent) is not independent. It indicates there should be an arrow between glucose and urea (and between urea and sodium). The following mediation analysis is the attempt to resolve this issue (results are shown in Table 6 from the CMAverse program [59]). From Table 6, the evidence for glucose → urea in one causal model is stronger than the urea → glucose in another model. Therefore, we decide that there should be an arrow pointing to urea from glucose. Similar from Table 6, the evidence for urea → sodium is stronger than sodium → urea. These results are incorporated in the causal model in Fig 3.

## Bringing in ferritin, creatine, and chloride into the causal model

From Table 4, there are only three variables besides the components of osmolality that are correlated with osmolality after conditional on age: ferritin, creatine, and chloride. If we consider sodium, urea, and glucose to be closer to osmolality, whereas ferritin, creatine and chloride

**Table 6. Mediation analysis with osmolality as the outcome variable.**

| X | M | Y | $p_M$ (CMAverse) (CI) | $p$-value |
|---|---|---|---|---|
| log(glucose) | log(urea) | log(osmo) | 0.22 (0.15 -0.29) | 8E-10 |
| log(urea) | log(glucose) | log(osmo) | 0.075 ( 0.047-0.11) | 2E-7 |
| log(urea) | log(sodium) | log(osmo) | 0.22 (0.16-0.28) | 2E-12 |
| log(sodium) | log(urea) | log(osmo) | 0.078 (0.05-0.1) | 2E-9 |

$p_M$: Proportion of the contribution in mediation path (indirect path) from X to Y through M, the 95% confidence interval; $p$-value: the $p$-value for testing $p_M = 0$.

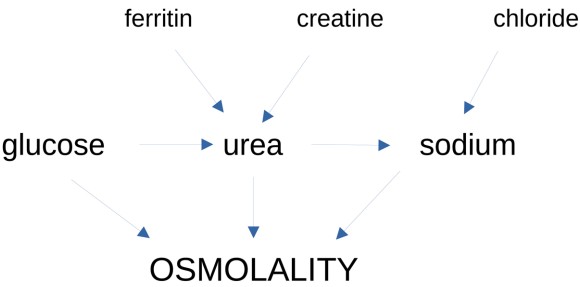

**Fig 3. Proposed model of relationship between osmolality and its three components (sodium, urea, glucose), as well as other associated quantities (chloride, creatine, ferritin).** Confounder-1 is a proposed confounder for creatine and urea (and probably ferritin). A candidate for confounder-1 is kidney dysfunction. Confounder-2 is a proposed confounder for sodium and chloride. A candidate for confounder-1 is salt level.

to be further from osmolality, as described in the pipeline in Fig 2, we continue to use correlation and mediation analysis to decide where and how to put these new variables into the causal model.

Table 7 shows the Spearman correlation and *p*-values between ferritin, creatine, and chloride with the three components of osmolality. If we select the component with the highest correlation to be paired with the new variable, then ferritin is paired with urea, creatine is paired with urea also, and chloride is paired with sodium.

Take ferritin and urea pair for example, one could use them and osmolality to do a mediation analysis, with two causal models. Of these two causal models, we can determine the causal arrow between ferritin and urea (the two already have causal arrows towards osmolality). The mediation analysis indicates that there is an arrow from ferritin to urea. This result is actually obvious as urea is a component of osmolality and should be closer to osmolality by definition. All our results are summarized in Fig 3.

## Discussions

Our dataset is imbalanced in gender, where the female to male ratio is almost 3 to 1. However, gender is not shown to be correlated with either mortality or hospitalization in this data. The Fisher test *p*-value for 2-by-2 count table formed by gender and mortality is 0.53. Similarly, the Fisher test *p*-value for gender and hospitalization is 0.92. As a result, the gender imbalance is not believed to cause any bias in our results. Furthermore, since our model reg4 adjusts both age and gender, even if there is a differential mortality/hospitalization risk in gender, it will be corrected in the reg4 model.

We have not collected a whole set of co-morbidities in this dataset, only the diabetes status. There are 71 patients (5.4%) labeled as diabetes, a relatively small sample size. Diabetic patients are significantly more likely to die ($12/71 = 17\%$ vs $18/1234 = 1.4\%$ for non-diabetic patients, *p*-value $= 9E-9$), or admitted to hospital ($20/71 = 28\%$ vs $110/1142 = 8.8\%$ for non-diabetic patients, *p*-value $= 5E-6$. How diabetes fit into the framework of high-osmolality causing COVID-19 severity will be a topic for future studies.

The non-traditional data analysis methods – collider analysis and mediation analysis – usually require three correlated variables. As long as there are multiple variables that are all pairwise correlated, these methods can be applied. We are not aware of previous attempts to apply these methods in the osmolality context, and our results should need independent check from other approaches. For collider analysis, it is assumed that confounding factors, if exist, do not impact the conclusion. Since it is difficult to assess this assumption, it is a limitation of the method. For mediation analysis, we only compare the relative (not absolute) strength of evidence for two causal models (in Table 6).

**Table 7**. **Correlation between one component of osmolality and one new variable.**

| pair | cor(Spearman) |
|---|---|
| glucose and ferritin | 0.32 (pv = 8E-32) |
| urea and ferritin | **0.41 (pv = 7.7E-54)** |
| sodium and ferritin | 0.15 (pv = 3.2E-8) |
| glucose and creatine | 0.2 (pv = 1.5E-13) |
| urea and creatine | **0.54 (pv = 8.5E-100)** |
| sodium and creatine | 0.1 (pv = 1.8E-4) |
| glucose and chloride | −0.13(pv = 5.8E-6) |
| urea and chloride | −0.068 (pv = 0.02) |
| sodium and chloride | **0.29 (pv = 3.1E-24)** |

Osmolality is not usually used as a prognostic marker. However, there have been isolated reports to support the idea of using osmolality to predict medical outcomes [37,67–70]. Using calculated osmolality as a prognostic biomarker for COVID-19 patients is even rare [39]. Even though each of the three components for calculated osmolality urea, glucose, and sodium, have been shown to be useful for predicting mortality from COVID-19 [71–76], the calculated osmolality has not been a focus of attention in hospital management of COVID-19 patients. The purpose of this paper is to show that osmolality itself is an excellent measure for COVID-19 prognosis purpose. In a sense, the combination of three good biomarkers should also be a good biomarker, but we point out the possibility that osmolality could be an even better biomarker than its individual components. Osmolality measure is particularly useful in distinguishing patients who are high in an individual component but not in other component, nevertheless, tend to not have metabolic problems.

The relevance of osmolality to disease severity might also be viewed from a physiological perspective. Osmolality combines status information from renal function, endocrine control, hydration balance, and metabolism. All these statuses will be disturbed during a multiple organ failure and systemic dysfunction. Although we only measured the osmolality at one time point, it could be interesting to monitor the trajectory of osmolality level. In some recent studies, the osmolality trajectories are determined for sepsis patients [77] or sepsis-associated encephalopathy patients [78], and those with increasing osmolality in time have the worst prognosis.

Although sodium, urea and glucose are three components of the calculated osmolality, it is usually ignored that these are not independent. By using mediation analysis, we ask the question of whether one of the components contribute to the calculated osmolality through other components indirectly. Our analysis shows that roughly 18% of the contribution from glucose is through urea, 22% of the contribution from urea is through sodium, and a small proportion, 7.8%, of the contribution from sodium is through urea.

To check whether there are other blood test measures to be associated with the calculated osmolality, even though these are not part of the official components, we calculate the non-parametric correlation (Spearman) conditional on age. The reason to condition on age is because age can be a common factor (confounder) to both a blood test measure and calculated osmolality [79] (see also Fig 1). This analysis led to only three other measurements: chloride, creatine, and ferritin.

Even though chloride is not part of the calculated osmolality in the formula we used, it is part of the measured osmolality [24]. Therefore, its association is not surprising. Our association analysis naturally paired chloride with sodium. It is not impossible that this association is due to the compound salt with both of them as components. Similarly, the creatine-urea could be reasonable as the two are both waste products of metabolic processes. Some kidney dysfunction could lead to a high level of both. Meanwhile, the link between ferritin and urea might be due to the role played by ferritin in regulating iron level and keeping kidney healthy.

A more severe disease leads to more damaged tissues, triggers more inflammation response. Some viruses infect lymphocytes and by increasing their destruction, it causes lymphopenia and a decrease in serum albumin, which is a negative acute-phase reactant, while it causes an increase in the levels of CRP and ferritin, which are positive acute-phase reactants. Published associations between ferritin and COVID-19 severity do not address the issue of causal direction [80,81].

We hope to compare our findings concerning osmolality-COVID19-severity association with other future data collected from other SARS-CoV-2 variants, as our data was mostly limited to the alpha-variant [45]. We hope our causal model for osmolality related variables in Fig 3 will be further examined by other lines of investigation in the future. We also hope that

future investigation of causal relation between osmolality and COVID-19 disease severity, either from a statistical causal inference or from biological studies, will provide hints for medical treatment. Furthermore, our work can be further improved by having more sample sizes and by collecting more potentially relevant confounders such as comorbidities.

## Author contributions

**Conceptualization:** Sirin Cetin, Ayse Ulgen, Hakan Sivgin, Wentian Li.

**Data curation:** Sirin Cetin.

**Formal analysis:** Ayse Ulgen, Meryem Cetin, Wentian Li.

**Investigation:** Ayse Ulgen, Hakan Sivgin, Meryem Cetin, Wentian Li.

**Methodology:** Sirin Cetin, Ayse Ulgen, Wentian Li.

**Project administration:** Hakan Sivgin.

**Resources:** Sirin Cetin, Hakan Sivgin.

**Software:** Ayse Ulgen, Wentian Li.

**Supervision:** Meryem Cetin, Wentian Li.

**Visualization:** Meryem Cetin.

**Writing – original draft:** Sirin Cetin, Ayse Ulgen, Wentian Li.

**Writing – review & editing:** Sirin Cetin, Ayse Ulgen, Hakan Sivgin, Meryem Cetin, Wentian Li.

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
