## [Decision Letter · Decision Letter 0]

22 Jul 2025

PONE-D-25-18731Osmolality as a strong predictor of COVID-19 mortality and its possible links to other biomarkersPLOS ONE

Dear Dr. Ulgen,

Thank you for submitting your manuscript to PLOS ONE. After careful consideration, we feel that it has merit but does not fully meet PLOS ONE’s publication criteria as it currently stands. Therefore, we invite you to submit a revised version of the manuscript that addresses the points raised during the review process.

We look forward to receiving your revised manuscript.

Kind regards,

Youhua Tan

Academic Editor

PLOS ONE

Journal Requirements:

Reviewers' comments:

Reviewer's Responses to Questions

**Comments to the Author**

1. Is the manuscript technically sound, and do the data support the conclusions?

Reviewer #1: Yes

Reviewer #2: Yes

2. Has the statistical analysis been performed appropriately and rigorously? 

Reviewer #1: Yes

Reviewer #2: Yes

3. Have the authors made all data underlying the findings in their manuscript fully available?

Reviewer #1: Yes

Reviewer #2: Yes

4. Is the manuscript presented in an intelligible fashion and written in standard English?

Reviewer #1: Yes

Reviewer #2: Yes

5. Review Comments to the Author

Reviewer #1: In Line 8, the author has mentioned as In our own work... it is not clear that the author wants to cite his previous paper or mention the present work. If it's present work, then it should not be in the introduction.

In line 17, the formula of osmolality is mentioned, which should be in methodology part

Line 72 to 81 should be in introduction part

Reviewer #2: Reviewer’s Comments:

The manuscript presents a comprehensive retrospective study investigating the predictive power of calculated plasma osmolality on COVID-19 mortality and hospitalization, incorporating extensive statistical analyses, including collider and mediation models. The study utilizes a substantial dataset (n = 1323) from Tokat, Turkey, and proposes a novel framework for integrating classical and causal inference methods in the interpretation of clinical biomarkers.

I suggest the following to the authors:

Major Strengths of the study

1. The investigation into osmolality as a prognostic marker is novel, as this parameter has been historically underutilized in COVID-19 clinical management despite being readily calculable from routine blood tests. The study highlights osmolality as a potentially stronger predictor of disease severity than its components (sodium, urea, glucose), which is an insightful and practical contribution.

2. The authors employ a multifaceted statistical approach, including logistic regression, ordinal logistic regression, collider analysis, and causal mediation analysis. This multi-tiered approach allows for a nuanced understanding of both direct and indirect biomarker associations.

3. The use of collider and mediation analyses is well-explained and innovative in the context of clinical biomarkers. The resulting causal model (Figure 3) is a valuable conceptual framework that can guide future research.

Suggestions for Improvement:

1. While the statistical rigor is appreciated, some of the results, particularly around transformed odds ratios and indirect effects, may be challenging for clinical readers to interpret. Inclusion of a summary table or visualization comparing biomarker performance (e.g., ROC curves or AUCs) would help translate findings into clinical utility.

2. The transformation of glucose leading to loss of significance warrants further discussion. Is this due to outliers, non-normal distribution, or a true change in association? A sensitivity analysis or data distribution plots might strengthen this point.

3. The exclusion of variables with high missingness (e.g., immune cell counts) is noted, but a summary on the extent and nature of missing data for other variables would be valuable, including any imputation strategy (if used).

4. The female-to-male ratio (approximately 3:1) is skewed. The authors should discuss how this imbalance might affect generalizability or statistical power in stratified analyses.

5. The clinical or physiological rationale linking osmolality (and its components) with COVID-19 severity is only briefly discussed. Further elaboration on how hydration status, renal function, or metabolic derangements contribute to poor prognosis would add depth.

6. While the authors have excluded several potential confounders (e.g., hematological malignancies, CKD), there is limited information on the patients’ comorbidities such as diabetes or hypertension, which may impact both osmolality and COVID-19 outcomes

Minor Comments:

• The manuscript could benefit from language polishing for grammar and flow in some sections (e.g., line 174–180).

• Consider providing actual effect sizes and confidence intervals for mediation paths in the main text rather than in supplementary tables only.

• It would be helpful to specify if the same causal framework can be generalized to other viral infections or critical illnesses beyond COVID-19.

This is a highly commendable study with methodological innovation and significant clinical relevance. The identification of osmolality as a composite yet potentially superior biomarker for COVID-19 prognosis is both timely and actionable. With minor revisions and improved clinical contextualization, this manuscript will make a strong contribution to the field of infectious disease biomarkers and statistical modeling in medicine.

Recommendation: Minor Revision

6. PLOS authors have the option to publish the peer review history of their article (what does this mean?). If published, this will include your full peer review and any attached files.

Reviewer #1: **Yes: **Richa panchgaur

Reviewer #2: No

---

## [Author Response · Author response to Decision Letter 1]

1 Aug 2025

Responses to Editor’s letter on Jul 22, 2025 (file also attached)

Sirin Cetin, Ayse Ulgen, Hakan Sivgin, Meryem Cetin, Wentian Li July 29, 2025

[AUTHORS] We would like to thank both reviewers for carefully reading the manuscript and providing helpful suggestions.

Reviewer 1:

In Line 8, the author has mentioned as “In our own work...”, it is not clear that the author wants to cite his previous paper or mention the present work. If it’s present work, then it should not be in the introduction.

[AUTHORS] It was our previous work published in another journal, Computational Biology and Chemistry; it is not about the current work in this submission. To avoid confusion, we have added “previous” in this sentence.

In line 17, the formula of osmolality is mentioned, which should be in method- ology part

[AUTHORS] see below.

Line 72 to 81 should be in introduction part

[AUTHORS] We did have the formula for osmolality in both the Introduction section and the Data/method section. The two instructions from the reviewer-1 are in conflict, but in the revision, we do remove the redundancy by putting this in the Introduction section (if the reviewer prefers putting this in the Method section, please let us know).

Reviewer 2:

The manuscript presents a comprehensive retrospective study investigating the predictive power of calculated plasma osmolality on COVID-19 mortality and hos- pitalization, incorporating extensive statistical analyses, including collider and me- diation models. The study utilizes a substantial dataset (n = 1323) from Tokat, Turkey, and proposes a novel framework for integrating classical and causal infer- ence methods in the interpretation of clinical biomarkers.

I suggest the following to the authors:

Major Strengths of the study

1. The investigation into osmolality as a prognostic marker is novel, as this parameter has been historically underutilized in COVID-19 clinical management despite being readily calculable from routine blood tests. The study highlights os- molality as a potentially stronger predictor of disease severity than its components (sodium, urea, glucose), which is an insightful and practical contribution.

2. The authors employ a multifaceted statistical approach, including logistic regression, ordinal logistic regression, collider analysis, and causal mediation anal- ysis. This multi-tiered approach allows for a nuanced understanding of both direct and indirect biomarker associations.

3. The use of collider and mediation analyses is well-explained and innovative in the context of clinical biomarkers. The resulting causal model (Figure 3) is a valuable conceptual framework that can guide future research.

[AUTHORS] We are very happy that our work is appreciated by the reviewer. Indeed, the under-utilization of osmolality in a clinical setting, and the introduction of simple causal inference in this application, are the two major contributions of this work. We thank the reviewer for his/her encouraging words.

Suggestions for Improvement:

1. While the statistical rigor is appreciated, some of the results, particularly around transformed odds ratios and indirect effects, may be challenging for clinical readers to interpret. Inclusion of a summary table or visualization comparing biomarker performance (e.g., ROC curves or AUCs) would help translate findings into clinical utility.

[AUTHORS] The reviewer’s comments made us re-examine not only our presentation of the methods, but also the appropriateness of some measurements, in particular the use of “transformed” odds-ratio (OR). We made substantial changes in presenting the results in Tables 1 and 2, and we offered a more detailed explanation for the methods used. Also, we adopted the reviewer’s suggestion to use the AUC as a measure of effect size.

Here is a list of changes made: (1) When “indirect path” is mentioned in the first time in Introduction, we have added that “using a jargon from the mediation analysis, to be explained in the Method section”. This would put the words in the framework of mediation analysis. (2) The AUC is calculated for Tables 1,2, and its definition is described, and meaning explained, in the Data/Methods section. (3) We no longer use odds-ratio (OR) as a measure to compare different factors, because we are unsure of an underlying assumption, i.e., a factor (or its log-

transform) follows a normal distribution. If some factors follow a normal distribution, some follow a log-normal distribution, and some do not follow either, then ORs are not compara- ble. Now, we provide a full explanation for OR for four versions of logistic regression in the Data/Methods section.

By the way, the previous normalization of OR may not be correct. Previously, OR for reg4 model is normalized by OR/exp(sd(log(x))). The correct way should be a normalization on log(OR): log(OR)/sd( log(x)). Towards this, we thank the reviewer for motivating us to improve the paper.

2. The transformation of glucose leading to loss of significance warrants further discussion. Is this due to outliers, non-normal distribution, or a true change in association? A sensitivity analysis or data distribution plots might strengthen this point.

[AUTHORS] The reviewer’s comment made us double check the raw data. We found out that one glucose value of 1.1 is too low to be true. Tracing back to the original source, we found out that this value should be 111. This error causes this deceased sample to have a very large negative value after log transformation, which single-handed made a significant signal insignificant (after log transformation). We have corrected this error and re-run all analysis with the updated data (Tables 1-7). We thank the reviewer for carefully reading our “footnote” sentences, and in turn, motivated us to find the ultimate cause of the problem.

3. The exclusion of variables with high missingness (e.g., immune cell counts) is noted, but a summary on the extent and nature of missing data for other variables would be valuable, including any imputation strategy (if used).

[AUTHORS] For immune cell counts, the missing rate is higher than 80% because some patients did not have the related blood test done.

These factors are also not included in the analysis due to higher missing rates: D-dimer (missing rate is 59%), activated partial thromboplastin time (APTT, 47%), fibrinogen (37%). For factors used, the extent of missing was already in column-2 of Table 1. Three factors have missing rates in the 9-11% range. For these patients, routine blood tests were done, but the missing data is not intentional, not pre-planned, and not systematic. In particular, it is not due to sample selection. The causes of the missing data could vary: it could be the variability

in COVID-19 admission protocols, clinical test priorities, or retrospective record gaps.

Importantly, the primary osmolality calculation in our study relied on serum sodium, potas- sium, glucose, and blood urea nitrogen, which were fully available for all participants. Thus, although chloride is part of electrolyte balance, its minor amount of missing data does not significantly impact osmolality estimation, especially given that the core osmolar components remained intact.

The information mentioned above are included in the revision. We did not carry out an imputation.

4. The female-to-male ratio (approximately 3:1) is skewed. The authors should discuss how this imbalance might affect generalizability or statistical power in stratified analyses.

[AUTHORS] The mortality and hospitalization rates are not correlated with gender: 2.7% male patients passed away, while the percentage is 2.1% for female; 9.6% male admitted to hospital, versus 9.9% for female. Fisher test of the 2-by-2 table (rows for gender, columns for mortality) leads to a p-value of 0.52, and that for hospitalization leads to a p-value of 0.92 (both are not significant). We also have not seen any significant signal from gender in other analyses. Therefore, the imbalance in gender in our samples does not seem to cause a problem. On the other hand, the regression model 4 adjusts gender in the analysis, so regression-4 result should remain true even if there is a gender effect. We have added a new paragraph in the Discussion section.

5. The clinical or physiological rationale linking osmolality (and its components) with COVID-19 severity is only briefly discussed. Further elaboration on how hydration status, renal function, or metabolic derangements contribute to poor prognosis would add depth.

[AUTHORS] We agree with the reviewer’s general assessment that osmolality incorporates measurements on several essential physiological functions, such as renal, endocrine, hydration balance, and metabolic situations. An abnormal osmolality level is an indication of multiple organ dysfunction, which is strongly associated with mortality. We have added a new para- graph about this in the Discussion section. Since we are not necessarily experts on physiology, and do not want to be too speculative, we did not write more.

6. While the authors have excluded several potential confounders (e.g., hema- tological malignancies, CKD), there is limited information on the patients’ comor- bidities such as diabetes or hypertension, which may impact both osmolality and COVID-19 outcomes

[AUTHORS] We do have diabetes information. The reviewer is absolutely correct that diabetic patients have a higher mortality rate than patients who do not have diabetes. We choose not to get into detail on this topic in this submission for the following reasons: (1) The diabetes rate in our dataset is low (∼ 5%), and the corresponding sample size for this subgroup is small. (2) osmolality is higher in diabetic patients than non-diabetic patients, which would lead to another multiple-variable-association situation. Currently, our causal analysis is limited to three factors at a time. When the number of pairwise correlated variables increases, we need

to think of an appropriate analysis strategy. A new paragraph has been added in the Discussion

section in the revision.

We do not have information on other morbidities.

Minor Comments:

* The manuscript could benefit from language polishing for grammar and flow in some sections (e.g., line 174–180).

[AUTHORS] The lines 174-175 before revision were: “To make different factors comparable, we divide the OR by esd[log(x)] (in the last column, as OR per ratio). If OR < 1, we use 1/OR instead in the calculation of the last column”.

As we decided not to use OR as a measure of effect size, but to use AUC instead, lines 174-175 are removed.

The lines 176-180 read: “We notice from Table 1 that conditional on gender and age generate makes the p-value larger (less significant), because some of the risk signal is shared with age variable (aging). Increasing age by one year increases the odd/risk of mortality by 16%, whereas increasing age by 16 years (one standard deviation of age in our data) raises the odd/risk by 10-fold (see Table 1).

This paragraph has been rewritten: “Table 1 also shows that for many factors, reg4 is less significant than reg1. One important difference between reg4 and reg1 is that reg4 adjusts the contribution from the age variable. One may treat age as a surrogate to deterioration of body at cellular and systemic level. On the other hand, the fact that even after an adjustment of age, many factors are still significant, indicates that the general aging process is not the only reason that some patients have higher risk for mortality.”

* Consider providing actual effect sizes and confidence intervals for mediation paths in the main text rather than in supplementary tables only.

[AUTHORS] The purpose of our mediation analysis is to compare two models: X1 → X2 →

Y and X2 → X1 → Y , and which one out of two has better data support (higher pM or smaller

p-value). The specific values are less important. This is the reason we only report pM (and its 05%CI) and p-value in Table 6 (by “supplementary table”, does the reviewer mean Table 6?

* It would be helpful to specify if the same causal framework can be generalized to other viral infections or critical illnesses beyond COVID-19.

[AUTHORS] Indeed, the type of causal analysis used in this paper, collider analysis and mediation analysis, involves three pairwise correlated variables. Therefore, as long as there are three variables that are all pairwise correlated, a similar strategy can be applied. We have added two sentences in the Discussion section to point this out.

This is a highly commendable study with methodological innovation and signif- icant clinical relevance. The identification of osmolality as a composite yet poten- tially superior biomarker for COVID-19 prognosis is both timely and actionable. With minor revisions and improved clinical contextualization, this manuscript will make a strong contribution to the field of infectious disease biomarkers and sta- tistical modeling in medicine.

[AUTHORS] Once again, we thank the reviewer for encouragements, as well as for carefully reading the submission and asking key questions which pushed us to greatly improve the paper.

---

## [Decision Letter · Decision Letter 1]

15 Aug 2025

Osmolality as a strong predictor of COVID-19 mortality and its possible links to other biomarkers

PONE-D-25-18731R1

Dear Dr. Ulgen,

We’re pleased to inform you that your manuscript has been judged scientifically suitable for publication and will be formally accepted for publication once it meets all outstanding technical requirements.

Kind regards,

Youhua Tan

Academic Editor

PLOS ONE

Additional Editor Comments (optional):

All the reviewers were satisfied with your addressing of their concerns/comments. I am thus pleased to accept your revised manuscript.

Reviewers' comments:

Reviewer's Responses to Questions

**Comments to the Author**

1. If the authors have adequately addressed your comments raised in a previous round of review and you feel that this manuscript is now acceptable for publication, you may indicate that here to bypass the “Comments to the Author” section, enter your conflict of interest statement in the “Confidential to Editor” section, and submit your "Accept" recommendation.

Reviewer #1: All comments have been addressed

Reviewer #2: All comments have been addressed

2. Is the manuscript technically sound, and do the data support the conclusions?

Reviewer #1: Yes

Reviewer #2: Yes

3. Has the statistical analysis been performed appropriately and rigorously? 

Reviewer #1: Yes

Reviewer #2: Yes

4. Have the authors made all data underlying the findings in their manuscript fully available?

Reviewer #1: Yes

Reviewer #2: Yes

5. Is the manuscript presented in an intelligible fashion and written in standard English?

Reviewer #1: Yes

Reviewer #2: Yes

6. Review Comments to the Author

Reviewer #1: The paper addressed the comments whatever it was previously mentioned. The journal can further proceed for pubication.

Reviewer #2: The manuscript titled "Osmolality as a Strong Predictor of COVID-19 Mortality and its Possible Links to Other Biomarkers" presents an important and timely investigation into the prognostic significance of osmolality in COVID-19 patients. The topic is highly relevant to current clinical practice, particularly for risk stratification and improving patient management during infectious disease outbreaks. The integration of osmolality with other biomarkers adds value to the existing body of literature and could potentially influence future diagnostic and therapeutic protocols.

7. PLOS authors have the option to publish the peer review history of their article (what does this mean?). If published, this will include your full peer review and any attached files.

Reviewer #1: No

Reviewer #2: No

---

## [Editor Report · Acceptance letter]

PONE-D-25-18731R1

PLOS ONE

Dear Dr. Ulgen,

I'm pleased to inform you that your manuscript has been deemed suitable for publication in PLOS ONE. Congratulations! Your manuscript is now being handed over to our production team.

Kind regards,

on behalf of

Dr. Youhua Tan

Academic Editor

PLOS ONE